



Climate
of the Past

# Droughts in Bern and Rouen from the 14th to the beginning of the 18th century derived from documentary evidence

**Chantal Camenisch**[1,2] **and Melanie Salvisberg**[1,2]

[1]Oeschger Centre for Climate Change Research, University of Bern, 3012 Bern, Switzerland
[2]Institute of History, Section of Economic, Social and Environmental History, University of Bern,
3012 Bern, Switzerland

**Correspondence:** Chantal Camenisch (chantal.camenisch@hist.unibe.ch)

**Abstract.** Droughts derive from a precipitation deficit and can also be temperature driven. They are dangerous natural hazards for human societies. Documentary data from the premodern and early modern times contain direct and indirect information on precipitation that allow for the production of reconstructions using historical climatology methods. For this study, two drought indices – the drought index of Bern (DIB) and the drought index of Rouen (DIR) – have been created on the basis of documentary data produced in Bern, Switzerland, and Rouen, France, respectively for the period from 1315 to 1715. These two indices have been compared to a third supra-regional drought index (SDI) for Switzerland, Germany, France, the Netherlands, and Belgium that was synthesised from precipitation reconstruction based on historical climatology. The results of this study show that the documentary data from Bern mainly contain summer droughts, whereas the data from Rouen rather allow for the reconstruction of spring droughts. The comparison of the three above-mentioned indices shows that the DIB and the DIR most probably do not contain all of the actual drought events; however, they detect droughts that do not appear in the SDI. This fact suggests that more documentary data from single locations, such as historical city archives, should be examined in the future and should be added to larger reconstructions in order to obtain more complete drought reconstructions.

# 1 Introduction

Droughts are threatening and dangerous natural hazards for human societies. They are complex phenomena that derive from a water deficit compared with normal conditions. They can also be also be temperature driven (van Loon et al., 2016). Depending on the nature/type of impact CE1, droughts can be classified into four different types: (1) meteorological drought, which describes a lack of precipitation over a certain period; (2) agricultural drought, which is a consequence of meteorological drought and affects the growth of crops; (3) hydrological drought, which usually occurs with a time lag following meteorological and agricultural drought and manifests as a shortage or lack of water in water courses, such as rivers and lakes; and (4) socio-economic drought, which occurs when a society suffers from the negative economic and social impacts of drought, such as during a subsistence crisis (Heim, 2002; Brázdil et al., 2018). Recent authors (van Loon et al., 2016; Brázdil et al., 2018; Metzger and Jacob-Rousseau, 2020) have emphasised the importance of interactions between the natural and human spheres in the occurrence and extension of droughts.

In order to understand present and future droughts, it is indispensable to investigate the past and to reconstruct historical droughts. This study aims to contribute to this target using the analysis of documentary evidence from two European cities – Bern, Switzerland, and Rouen in Normandy, France – for the period between 1315 and 1715 by answering the following questions:

– What is the potential of municipal documentary data for the reconstruction of drought in pre-instrumental periods?

– Which drought years or drought seasons can be found in the documentary data of the two cities?

– How do these drought reconstructions compare to a supra-regional drought index?

For this purpose, a separate drought index was created for each city. As the number of detected droughts is limited for the two locations in the pre-modern period (due to the characteristics of the sources), a third supra-regional drought index for the area of modern Switzerland, France, the Netherlands, and Germany was developed using a couple of pre-existing precipitation reconstructions or catalogues with descriptions. As droughts belong to the most prominent natural hazards, increasing attention has been paid to this topic in recent years. Different approaches to exploring tree-ring data and other archives of nature allow for the reconstruction of past droughts (e.g. van der Schrier et al., 2007, 2013; Briffa et al., 2009; Helama et al., 2009; Büntgen et al., 2010a, b; Todd et al., 2013; Neukom et al., 2014; Nash et al., 2016a; van Loon et al., 2016; Haslinger and Blöschl, 2017; Seftigen et al., 2017; Dobrovolný et al., 2018; Ljungqvist et al., 2019). Moreover, the Old World Drought Atlas (OWDA) provides a very comprehensive and easily accessible overview of drought in Europe over the last 2 millennia (Cook et al., 2015).

Apart from tree-ring analyses, documentary data provide another approach to drought reconstruction of the past. For instance, early instrumental precipitation measurements – often in combination with other documentary data (Hannaford et al., 2015; Brázdil et al., 2019; Erfurt et al., 2019; Harvey-Fishenden et al., 2019) – can be used for the creation of drought series. Moreover, documentary data also form the basis for the reconstruction of individual events (e.g. Dodds et al., 2009; Brázdil et al., 2013; Wetter et al., 2014, Nash et al., 2016b; Kiss, 2017; Camenisch et al., 2020). For periods and regions where no such early measurements are available, climate indices can be constructed using historical climatology methods (e.g. Brázdil et al., 2013, 2016; Možný et al., 2016; Garnier, 2018).

In most cases, the reconstructions presented here either focus on a whole region or primarily use one source sample as their basis. In this study, the focus lies on two cities – Bern and Rouen – and the documentary data produced there that are known for their satisfying source density. This paper is structured into six sections. After the introduction (Sect. 1), the data are presented in detail (Sect. 2). The applied methods are then discussed in Sect. 3, before the results are presented (Sect. 4). Finally, a discussion of the results (Sect. 5) and a short conclusion (Sect. 6) follow.

## 2    Data

In regard to late medieval and early modern times, weather-sensitive documentary data usually encompass at least narrative sources such as chronicles, annals, and other historiographic texts, as well as administrative sources like account books and ship logs; moreover, pamphlets as well as other early printed media and early weather diaries (Pfister et al., 1999; Camenisch, 2015a; Garnier, 2018) are included. In general, documentary data contain either direct information or indirect (proxy) information on weather and weather impacts on human societies. In the case of direct information, this means, for instance, descriptions of weather conditions over a certain period. Depending on the interests of the observer, the sources focus either on descriptions of extreme weather events and natural hazards or, in fortuitous cases, even on serial accounts of weather descriptions that include average weather conditions (Pfister, 1999; Camenisch, 2015b). If sources contain series of proxy data – which give indirect information on weather-related processes such as plant or ice phenology – further statistical analyses are necessary (e.g. Pribyl et al., 2012; Labbé et al., 2019). As the characteristics of the sources are different, it is necessary to consider the potential and the limits of the weather-related piece of information for each source and, in some cases, even for each record (Camenisch, 2015b).

In this study, two different samples of historical sources that concern either the city of Bern, in today's Switzerland, or the city of Rouen in Normandy, France, are analysed. The reason for this choice is to examine how a comprehensive source sample from a single location – or, in this case, two single locations – can be of use for drought reconstruction in pre-modern and early modern times. In the period from 1315 to 1715, Bern was an important city situated in the western part of the Swiss Plateau. During the 15th and 16th centuries, the city and republic of Bern was able to expand its political and territorial power and establish the largest city state north of the Alps (Hesse, 2003; Camenisch, 2019). From the 14th to the 18th century, Rouen was a large and flourishing city in Normandy. The city was significant for the political administration of Normandy as well as for the trade of goods between Paris and the Norman coast (Mollat, 1979). However, over the course of the 400 years examined here, the fates of Bern and Rouen changed from time to time, and both cities suffered war, epidemics, and famines during certain periods. Both cities possess historical archives with extensive documentary data that have different origins, contexts, and purposes. In the case of Bern, narrative sources such as the chronicles of Diebold Schilling (Schilling, 1985) or Johann Haller and Abraham Müslin (Haller and Müslin, 1800) were primarily analysed, along with the proceedings of the city council – the so-called "Ratsmanuale" – and further sources from the cities' administration. For the earliest years, sources from nearby areas were also taken into consideration. Moreover, drought-related entries from the Euro-Climhist climate

database (Pfister and Rohr, 2015) were added to the source sample.

For the drought reconstruction in Rouen, narrative and historiographic sources were once again examined, such as the chronicle of Pierre Cochon (Cochon, 1870) and the "délibérations" – the proceedings of the city council of Rouen – in addition to other administrative sources. The reason that these specific sources were chosen was that at least the proceedings of the council of both cities cover a large part of the investigated period. Municipal accounts were not part of this analysis because there are more accounts than could be analysed in a reasonable time for Bern, and there are too many gaps in the accounts for Rouen.

These sources mainly contain descriptions of extreme weather events and no frequent or serial proxies. As droughts are such extreme weather events, this is no disadvantage at all. However, as a consequence, only events remarkable to the contemporary authors or events that provoked action from the city government appear in these sources. This means that it is likely that not all droughts can be detected using these sources.

The descriptions in the sources are often detailed and contain information regarding the weather impacts on nature, agriculture, and the economy, as the following example from Cochon (1870) shows: "And in this year 1422, there was a general abundance of all goods, such as grain. There was good and strong wine, which was wonderful, and likewise an abundance of all fruit. The summer was so dry that the good people living in village in elevated altitudes had to come down to the rivers in order to get drinking water. This weather lasted until mid-October.".[1] However, shorter descriptions also exist that focus only on the weather conditions of a certain period, such as that from Schwinkhart (1941): "In the aforesaid year (1517), a dry and hot summer and autumn occurred from the beginning onwards until the end of the last autumn month (November).".[2] The second example also shows that the authors tend to mention drought and heat together – at least for the spring, summer, and autumn seasons. This is the reason why it is not always easy to distinguish whether the lack of precipitation or the elevated temperatures were responsible for the impacts. In regard to the above-mentioned four drought types, all four types can appear in the sources, as long as they were remarkable enough.

As argued above, it can be anticipated that not all droughts in Bern and Rouen could be detected with this source sample. In order to compare the two city-based series with a larger area, a third index was created. The data for this supra-regional index were derived from other pre-existing climate reconstructions that are primarily based on documentary data rather similar to the previously described sources (Buisman, 1996, 1998, 2000, 2006; Schwarz-Zanetti, 1998; Pfister, 1998, 1999; Glaser, 2013; Le Roy Ladurie, 2004).

## 3 Methods

The characteristics of the documentary data used for this study suggest the creation of unweighted drought indices using historical climatology methods. In this research field, creating indices has a long tradition. Pfister et al. (2018) proposed separate temperature and precipitation indices using a seven-degree scale (see Table 1); Brázdil et al. (2013) used all three dry-index values in their drought reconstruction, which, of course, gives a comprehensive picture.

However, as the sources used here show a tendency toward extreme events, only the extremely dry (−3) and very dry (−2) index points were used for the reconstruction. The less frequent −1 events were not included in this reconstruction because they would represent a lack of precipitation of less than a few weeks and are, therefore, not understood as drought. In a first step, the sources were searched for drought descriptions. In a second step, these drought descriptions underwent a qualitative analysis, which meant that the droughts were classified by comparison according to their intensity and duration (Pfister, 1999; Pfister et al., 2018; Glaser, 2013; Camenisch, 2015a and b). To attribute an index value of −2 to a drought, at least 1.5 months of reduced precipitation had to have been described. These very dry weather conditions could have had either negative or positive impacts, such as a poor or good harvest respectively. Extremely dry (−3) periods lasted at least 2 months, and the impacts were described as negative, such as a lack of drinking water or harvest failure. Whenever possible, the droughts were attributed to the meteorological season. This has been done for the drought index of Bern (DIB) and the drought index of Rouen (DIR).

The third index – a synthesised drought index (SDI) for the area of Switzerland, Germany, France, the Netherlands, and Belgium – was created on the basis of pre-existing precipitation reconstructions (Buisman, 1996, 1998, 2000, 2006; Schwarz-Zanetti, 1998; Pfister, 1998, 1999; Glaser, 2013; Le Roy Ladurie, 2004). Pfister and Schwarz-Zanetti used the same seven-degree scale as that presented in the DIB and the DIR. Buisman and van Engelen used a four-degree index for precipitation and classified an extremely dry season as "--". In the cases of Buisman (1996, 1998, 2000, 2006), Glaser (2013), and Le Roy Ladurie (2004), the descriptions of the different seasons were also taken into account. Because the area of Switzerland, Germany, France, the Netherlands, and Belgium is very large and precipitation patterns often have a smaller scale with very different impacts, the SDI does not contain a further classification apart from the −2 and −3 in-

---

[1] "Et, en celle année .cccc. xxij., fu tant habundanche de tous bienz universelement, tant blés, vinz si bonz et si fors que c'estoit grant merveille, et semblablement de touz fruytages. Et fist si sec, cet esté, que les bonnes genz des haux villagez ne povoient avoir point d'eaue, s'il n'allassent ès rivierez. Et ainssy se passa le temps jusques en la my octobre."

[2] "In dem obgenanten jare ward ein drochner vnd heyßer summer vnd herbst von anfang bys zů ende des letsten herbstmonat."

https://doi.org/10.5194/cp-16-1-2020

**Table 1.** Seven-degree precipitation index (Pfister, 1999; Brázdil et al., 2013; Glaser, 2013).

| −3 | −2 | −1 | 0 | 1 | 2 | 3 |
|---|---|---|---|---|---|---|
| Extremely dry | Very dry | Dry | Normal | Wet | Very wet | Extremely wet |

dex values; moreover, it does not include seasonal division. A drought appears in the SDI only when at least 2 months were declared as very dry (−2) or extremely dry (−3) and when more than one region was affected. This simplification was necessary due to the different approaches chosen in the used reconstructions and owing to the complex documentary data that form their basis.

## 4 Results

The DIB shows droughts over the whole 400-year analysis period (see Fig. 1). Most of the droughts were classified as very dry (−2), and only three extremely dry (−3) periods in the years 1462, 1473, and 1540 were described in the Bernese documentary data. There are certain accumulations of drought periods at the end of the 14th century, in the second half of the 15th century, in the middle of the 16th century, and during the 1670s and early 1680s. Repeated drought periods also appear in the DIR, but their number is smaller than in Bern. Again, most of them belong to the very dry (−2) category, and extremely dry periods were only reported in the years 1624, 1625, and 1684. Accumulations of droughts are only visible in the middle of the 16th century, during the 1620s, and during the 1670s and 1680s. There are a few years when both the DIB and the DIR show drought during the same period: 1363, 1546, 1556, 1567, and 1681.

In regard to the seasonal distribution of the droughts (see Fig. 2), it can be stated that more is known about the seasonality in Bern than in Rouen. In Bern, the descriptions of summer droughts clearly prevail, whereas more spring droughts are reported in Rouen. The reason for this is that repeated rogation ceremonies for the end of a drought are described at the end of May or the beginning of June in the records for Rouen. As no other information on precipitation is available for most of those years in Rouen, only a spring drought can appear in the reconstruction – even if it is very possible that the drought did not end with the rogation ceremony. We can only guess as to why no rogation ceremonies are reported for late summer or other seasons, but it is probable that the date of such religious practices in Rouen is linked to the phenology of cultural plants growing in the areas around the city. Drought reports for the winter and autumn seasons are scarcer in both locations, but a few examples were found in both Bern and Rouen.

As the SID comprises a much larger area and more independent documentary data, more droughts clearly appear in this synthesised reconstruction (see Fig. 3). It is important to note that the reconstructed droughts do not concern the whole area in most cases – the indicated years only show when drought periods are reported in at least two of the examined regions. However, a cluster of repeated extreme droughts is identifiable in the first half of the 16th century and again in the first half of the 17th century. Droughts only appear in all three indices in 3 years: 1556, 1567, and 1681. Between the DIR and the SDI, 4 years match: 1422, 1624, 1678, and 1684. However, more similarities clearly exist between the DIB and the SDI, because simultaneous drought reports are available for 13 years: 1385, 1393, 1442, 1462, 1471, 1472, 1473, 1517, 1540, 1546, 1558, 1561, and 1676.

The comparison with the SDI also shows that not all droughts are reported in the two sets of documentary data examined. The SDI contains years when drought was reported somewhere in Switzerland, Germany, France, the Netherlands, and Belgium; however, this does not mean that drought affected the whole area in those years. Therefore, the number of droughts reconstructed in the SDI is certainly higher than the actual occurrence of drought at a single location, and the number is also higher than in the DIB and DIR. Among the drought years in the DIB and DIR with no corresponding counterpart in the SDI were 1333, 1363, 1382, 1549, 1560, 1586, and 1680; 1555 and 1583 in the DIR had no corresponding counterpart in the SDI CE2.

More detailed descriptions are presented below for the years that were reported as very dry or extremely dry in all three reconstructions. These are not necessarily the years with the highest number of source descriptions. Many more descriptions are available for the years 1473 and 1540 (Wetter et al., 2014; Camenisch et al., 2020), but those years only appear in the DIB and the SDI.

### 4.1 The year 1556

A Bernese source tells us that a very hot and dry summer occurred in the year 1556. A great quantity of good wine was available for low prices, whereas there was not much grain. This was the reason that grain prices increased, and from mid-November high prices had to be paid. In the Bernese highlands a cattle murrain raged (Haller and Müslin, 1800). In Rouen, apart from the short mention of a drought and heat period, long descriptions of governmentally organised cleaning of the streets and the river Seine appear in the proceedings of the city council. The city council also specifically prohibited throwing any garbage into the streets or into the river throughout the summer, because it could disturb river traffic.

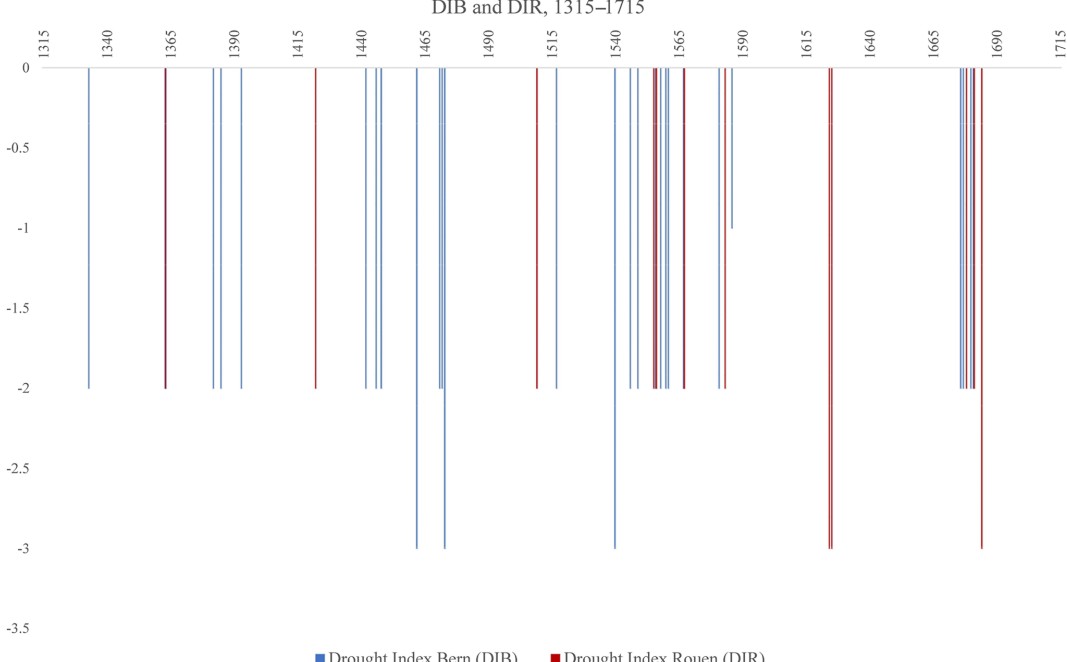

**Figure 1.** Drought index of Bern (DIB) and drought index of Rouen (DIR), 1315–1715.

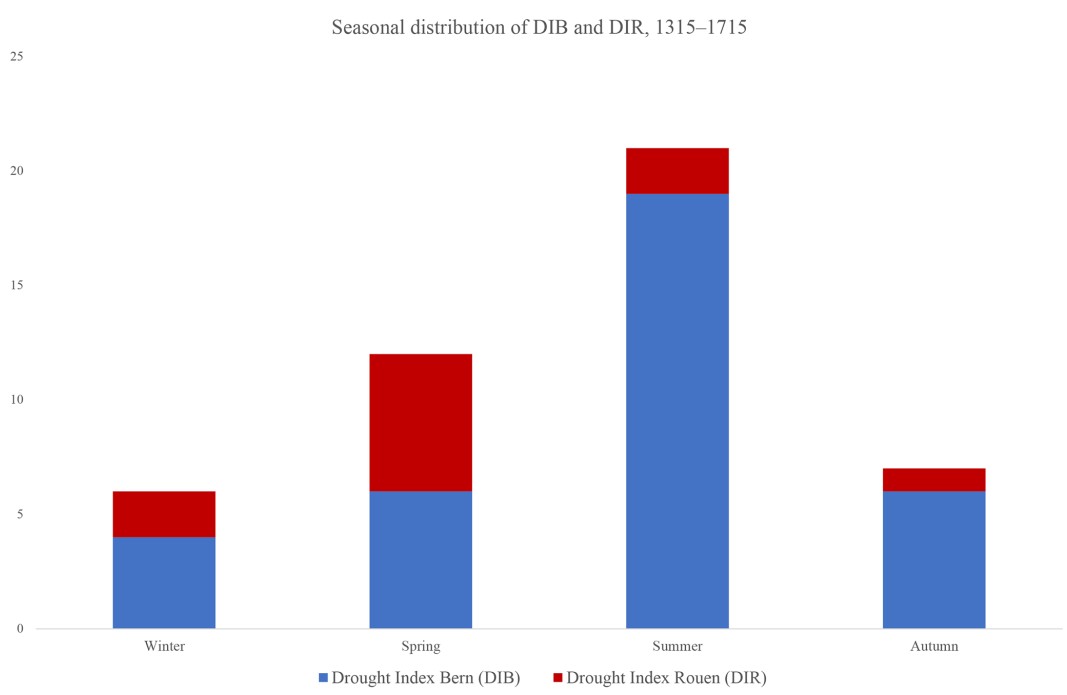

**Figure 2.** Seasonal distribution of drought index of Bern (DIB) and drought index of Rouen (DIR), 1315–1715.

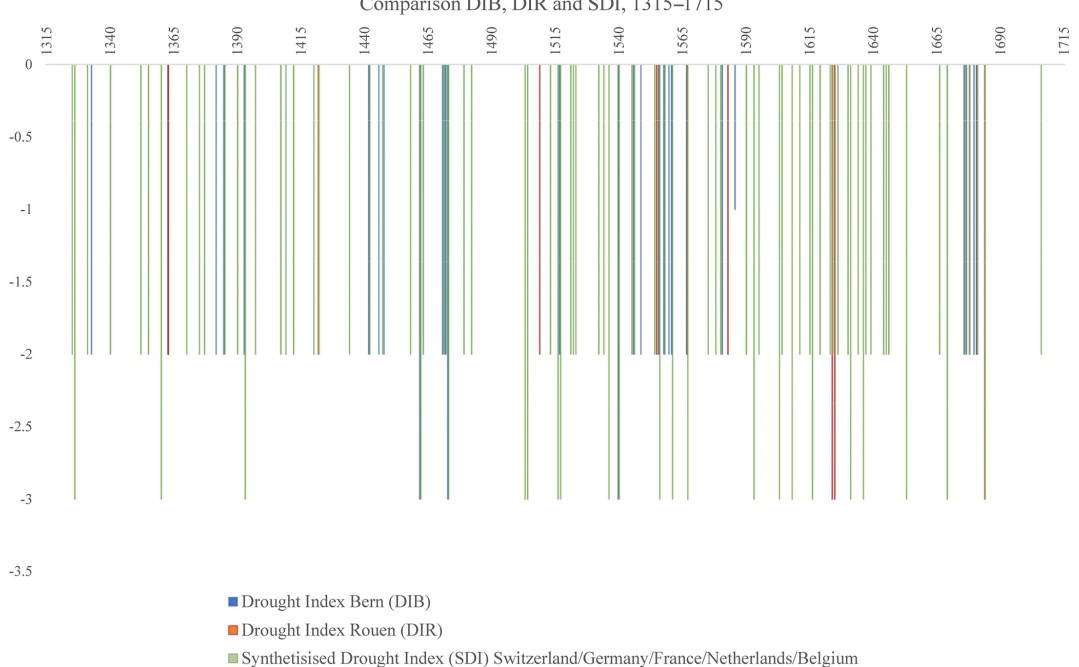

**Figure 3.** Comparison of the drought index of Bern (DIB) and the drought index of Rouen (DIR) with the synthesised drought index (SDI) for Switzerland, Germany, France, the Netherlands, and Belgium (1315–1715).

In the meantime, Rouen suffered from a terrible epidemic disease, which was linked to the heat in the eyes of contemporaries (Délibérations de la ville de Rouen, 1556). In southern Germany, the drought had already started in spring, and the extreme lack of precipitation affected the whole area over the course of the summer. The drought lasted until at least November and was only locally interrupted by downpours of rain and the following floods at the beginning of July. France was also hit by extreme heat and drought. The consequences of these weather conditions were a very early beginning to the vintage (grape harvest) CE3 as well as a poor harvest (due to the lack of water), low water levels, and the drying up of brooks and swamps (Glaser, 2013; Le Roy Ladurie, 2004).

## 4.2   The year 1567

In 1567, it was again Haller who described a dry but cold late spring (Haller and Müslin, 1800). As a consequence, the hay harvest was poor in the area of Bern. In Rouen, not much information is given about the weather conditions apart from a note that at the end of May stating that the archiepiscopal administration organised a rogation in order to end the drought (Délibérations du chapitre de la cathédrale de Rouen, 1567). Such a procession is held only when the problems resulting from a drought are already severe. Spring, summer, and autumn were also clearly too dry in Germany, which led to an above-average vintage. As a consequence of the drought, wildfires raged in the Thuringian forest and the Harz.

## 4.3   The year 1681

For the year 1681, the Euro-Climhist database provides us with several pieces of information about little or no precipitation in the area of Bern in late spring and summer (Pfister and Rohr, 2015). At the beginning of June, a rogation ceremony was held by the city government of Rouen. The reason for this was a drought period, as described in the council proceedings (Délibérations de la ville de Rouen, 1681). As Le Roy Ladurie (2004) described, other parts of France also suffered from this dry spell, although the grain prices remained at a moderate level. In Germany, spring was already very dry, and rain showers only occurred on a few occasions in summer and never with a sufficient amount of water. These weather conditions did not change before autumn, when a wet and unsettled period started (Glaser, 2013).

The comparison between the DIB and the DIR raises the question of why such a diverging number of droughts is possible in two rather similar sets of documentary data. The answer reveals a deeper insight into the characteristics of the source types used and into the potential of these municipal documentary data. Both data sets contain administrative sources, such as the proceedings of the city council and narrative sources. Both source types report only very dry or extremely dry periods. In regard to the city council proceedings, this was only the case when the councils were forced to react to the negative impacts of (extreme) droughts, whereas the narrative sources usually focus on outstanding events in general. It also seems that the narrative sources in Bern are

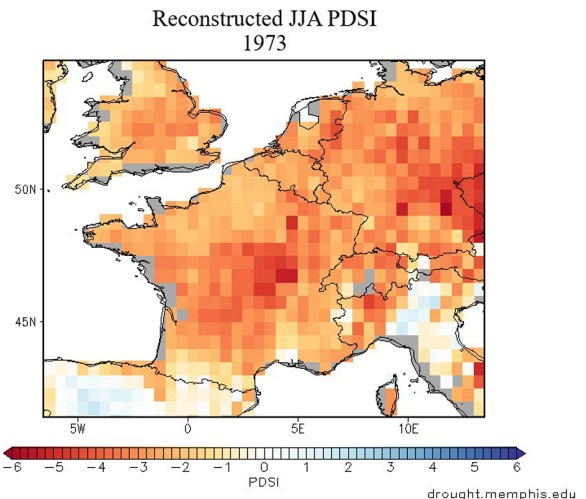
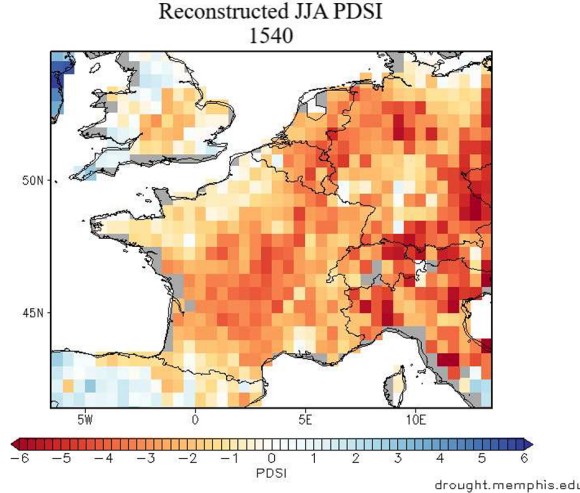

**Figure 4. (a)** Map of the year 1473 from the OWDA. **(b)** Map of the year 1540 from the OWDA (Cook et al., 2015).

more drought sensitive than those in Rouen. Furthermore, more narrative sources are available for Bern. This is most probably the reason for the different frequency of drought descriptions.

## 5 Discussion

The information from the DIB and the DIR presented here show droughts in Bern and Rouen for a period of 400 years, and there are similarities between these two indices as well as in comparison to the SDI. Nonetheless, it is striking that extreme drought years such as 1473 and 1540 are only reported in Bern but not in Rouen. In the case of 1473, there is a gap in the proceedings of the city council in Rouen, and the other examined sources do not mention the weather conditions in this year. However, the OWDA shows that this drought also affected Normandy (see Fig. 4a). The case of 1540 is a bit different. In this year, the source density seems sufficient, but the lack of precipitation may not have been severe enough in Normandy to provoke any reaction from the city government. The OWDA also suggests more precipitation in Normandy than in other European regions in that year (see Fig. 4b). These examples of two extreme drought years demonstrate that, when sources such as these proceedings of the city council are examined, it is possible that even major events are not recorded. For this reason, it is advisable to investigate several locations in order to minimise errors and gaps.

Moreover, a quantitative comparison of the DIB, DIR, and SDI with the OWDA for approximately the same area as was examined for the SDI (45.09 to 53.92° N, −3.91 to −15.23° E CE4) was carried out for this research. Of course, the DIB, DIR, and SDI series reconstructed here only show extreme events, whereas the OWDA is a continuous series; thus, the quality and characteristics of the data are quite

different. However, Pearson correlations show certain but moderate similarities between the DIB and the OWDA ($r = 0.32$), the DIR and the OWDA ($r = 0.22$), and the SDI and the OWDA ($r = 0.42$) – although only the latter result is statistically significant.

A direct comparison of very dry years in the OWDA with the reconstructions presented here also shows matching years. For this purpose, all values of the Palmer drought severity index (PDSI) applied in the OWDA of −2.5 or less were taken into account, which includes the drier half of moderate droughts as well as the severe and extreme droughts. Most of the matching years can be found in the comparison of the OWDA with the SDI: 1325, 1326, 1360, 1385, 1397, 1420, 1422, 1434, 1442, 1447, 1462, 1473, 1504, 1517, 1540, 1590, 1603, 1616, 1624, 1636, 1644, 1653, 1666, 1676, 1681, and 1684. Moreover, the comparison of the OWDA with the DIB still shows a number of matching years (1385, 1442, 1462, 1473, 1517, 1540, 1676, and 1681), whereas the DIR and the OWDA only share a few common drought years (1422, 1624, 1681, and 1684). In some cases, such as 1371, 1384, 1394 1464, 1461, 1525, and 1635, droughts in the DIB, DIR, or SDI occurred 1 year earlier or later than in the OWDA. The reason for this could be an uncertainty in the dating of the documentary data. In cases when the information is derived from city councils proceedings, dating errors are unlikely. However, if the data are taken from chronicles and other retrospective narrative texts, uncertainty in dating should be considered. Other years do not show a congruence. In these cases, one must recall that the tree-ring-based OWDA mainly shows spring and summer droughts, whereas the DIB and DIR also contain autumn and winter droughts.

## 6    Conclusions

The DIB and DIR indices presented here show drought reconstructions of two data sets of municipal documentary data over 400 years. On the basis of the information given in the sources, it was possible to distinguish between very dry and extremely dry periods by comparing the descriptions of the weather conditions and the impacts on society. Although the sources have a tendency to report outstanding and extreme events, in both the cities of Bern and Rouen very dry periods clearly occur more frequently than extremely dry periods do. In many cases, the droughts could be attributed to certain seasons. In both reconstructions, droughts mostly appear in spring and summer. A comparison to a third drought index for the areas of Switzerland, Germany, France, the Netherlands, and Belgium shows that most (although probably not all) droughts could be detected in the sources of Bern and Rouen. However, this also shows that the DIB and the DIR contain droughts that are not known to the SDI. A last comparison to the OWDA reveals that many of the drought years reported in Bern and/or Rouen were also visible in tree-ring-based reconstructions. This means that the analysis of geographically limited source samples, such as data sets of municipal sources, can contribute to a more detailed understanding of the extent and severity of droughts. In some cases, these types of data sets even have the ability to detect currently unknown droughts. For future research, investigators should consider whether more drought reconstructions for single locations with a sufficient data density can be made and whether they can be linked to a larger grid of drought reconstructions.

**Data availability.** All data used in this paper are included in the Supplement. CE5

**Supplement.** The supplement related to this article is available online at: https://doi.org/10.5194/cp-16-1-2020-supplement.

**Author contributions.** CC undertook research, carried out the analysis and wrote the paper. MS supported CC with respect to writing the paper, analysing the data and developing the research approaches. CE6

**Competing interests.** The authors declare that they have no conflict of interest.

**Special issue statement.** This article is part of the special issue "Droughts over centuries: what can documentary evidence tell us about drought variability, severity and human responses?". It is not associated with a conference.

**Acknowledgements.** Chantal Camenisch acknowledges support from the Swiss National Science Foundation through an "Advanced Postdoc.Mobility" grant, the Burgergemeinde Bern, and the Berne University Research Foundation. The paper is a contribution to the PAGES (Past Global Changes) Climate Reconstruction and Impacts from the Archives of Societies (CRIAS) working group.

**Financial support.** This research has been supported by the Swiss National Science Foundation (grant no. P300P1_171551). TS1

**Review statement.** This paper was edited by Jürg Luterbacher and reviewed by Fredrik Charpentier Ljungqvist and one anonymous referee.

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

## Remarks from the language copy-editor

CE1     Please note the change.
CE2     Please confirm the change based on your feedback. If this is incorrect, please provide me with a corrected version of this sentence. Thank you.
CE3     Please note the addition.
CE4     Please note the edit. The dash looked strange here because of the minus sign. I have reformatted the text using words.
CE5     Please note the change.
CE6     Please confirm the change.

## Remarks from the typesetter

TS1     Please note that the funding information has been added to this paper. Please check if it is correct. Please also double-check your acknowledgements to see whether repeated information can be removed or changed accordingly. Thanks.
TS2     Please provide volume.
TS3     Please provide volume.
TS4     Please confirm.