# Peer review of "Droughts in Bern and in Rouen from the 14th to the beginning of the 18th century derived from documentary evidence"

_Climate of the Past, 2019_

## Referee Comment (RC1) · Anonymous Referee #1 · 25 Oct 2019

Camenisch and Salvisberg present a new documentary-based comparison of droughts in Bern and Rouen from the 14th century to the 18th century. This effort is worth to publish in itself although the article suffers from a number of deficiencies. As I still think that the article is worth to publish, I will only highlight my points of criticism:

1) The authors are missing a number of recent important studies regarding historical droughts in Europe. While they are pretty well aware of the research in historical climatology, they are missing works in other fields of high-resolution palaeoclimate science.

2) I am lacking a quantitative comparison between the new documentary-based drought reconstructions and the tree-ring based Old World Drought Atlas. Without

such a quantitative comparison it is very hard – or even impossible – to really know the nature of the similarities and disagreements.

3) Several of the figures are totally unreadable (and unpublishable). For example, the years are overlapping with each other so it cannot be read: much longer increments (say, 25 years) are needed for a clearer visualization. I also recommend the authors to look at articles in Climate of the Past for getting inspiration how to improve the graphs.

Minor comments: Line 7: Droughts can also be temperature-driven without a decrease in precipitation. I also would consider it an overstatement that droughts belong to the most dangerous natural hazards. Lines 40–42: Add citations to Seftigen et al. 2017, Helama et al. (2009) and Ljungqvist et al. (2019, ERL) here.

---

## Referee Comment (RC2) · Anonymous Referee #2 · 6 Jan 2020

The paper is well-written, logically built up. The general argumentation, introduction, data and methodology chapters are presented rather properly. I have a few comments to the Results and the Discussion parts of the paper. Major comments: 1) I miss the general analysis and direct comparison of the two series in the Results chapter. Here only the greatest extremes are addressed and – despite the fact that the authors have made significant efforts to build up large-scale comparative series from Western Europe – only the greatest large-scale extremes are addressed in brief. Here, I think, a proper comparative analysis should be added: how different the individual series are from the regional series, what could be the reasons (e.g. source availability? Scale of events? Differences in the intensities or impacts on societies? etc.) for these differences (and the similarities). 2) The authors provide direct comparisons of the evidence derived from the two cities in the Discussion chapter. As the entire paper (so as the title) is mainly concentrated on this topic, in my opinion this part of the Discussion chapter should be moved to the Results chapter. 3) I think the Discussion chapter could be organised somewhat differently: here smaller but important specific topics could be discussed. I think the discussion of 1473 and 1540, and its different appearance in the two cities are a good idea for one topic (i.e. I would keep it there), but plenty of other important questions could be addressed here. For example, the authors refer to the tree-ring based OWDA as one of the applied databases in the paper: in the Discussion chapter the authors could e.g. systematically compare the OWDA with the documentary evidence and list similarities and potential differences. Other possibility could be, for example, the discussion of uncertainties.

Minor points: 1) It is rather remarkable that in Rouen only the droughts prior to summer could be detected. In the paper the authors explain this situation with source availability. I was just wondering: is it possible that for such a large town as Rouen no source exist at all that describe any other part of the year that contains any weather-related information? No any weather(-related) information at all in narratives, no other institutional documentation (e.g. municipal accounts)? It is rather unusual, especially with regards to the later part of the study period – and if this is the case, I think, should be more emphasised in the paper – already in the Source description part. The authors explain this phenomenon mainly with the difference in documentation practice. However, documentation practice is always related to 2) Is it really the case that in the documentation of the two cities only and exclusively the great and extreme dry conditions are mentioned, and never even moderate dry conditions? It is true that, usually, references on moderate dry conditions in European documentation are less pronounced. Still, they appear in documentation. Thus, it is a rather interesting and unique fact that, as the authors suggested, in neither of the two cities any "dry conditions" (i.e. without referring to any extreme) have been mentioned. Or, do you mean that all cases when "dry conditions" were mentioned had to be great (i.e. no. 2) or

extreme (no. 3) droughts? It is really just a question out of curiosity. The question is also addressed to understand better the level of potential uncertainties of the index values. 3) The authors suggest that mainly spring and summer droughts could be detected. This is a typical characteristics in Western and Central Europe (actually, also in Eastern Europe). Does this mean that the authors found no autumn and winter drought mention at all? Or did you find some? Because if you did, it would be perhaps also interesting for a short discussion in the "Discussion" chapter. Maybe not – it is up to you (just for a further potential idea into the Discussion chapter).

---

## Author Comment (AC1) · 7 Feb 2020

Many thanks for your kind and helpful comments, we appreciate them very much.

Referee #1: 1) The authors are missing a number of recent important studies regarding historical droughts in Europe. While they are pretty well aware of the research in historical climatology, they are missing works in other fields of high-resolution palaeoclimate science.

*Response: Many thanks for this useful and necessary comment. We open the focus of our state of the art to other fields and add the respective recent publications on

drought reconstruction.

Referee #1: 2) I am lacking a quantitative comparison between the new documentary-based drought reconstructions and the tree-ring based Old World Drought Atlas. Without such a quantitative comparison it is very hard – or even impossible – to really know the nature of the similarities and disagreements.

*Response: Many thanks also for this very useful comment. Of course, we agree and add such quantitative comparisons.

Referee #1: 3) Several of the figures are totally unreadable (and unpublishable). For example, the years are overlapping with each other so it cannot be read: much longer increments (say, 25 years) are needed for a clearer visualization. I also recommend the authors to look at articles in Climate of the Past for getting inspiration how to improve the graphs.

*Response: Many thanks for these remarks. We will improve the figures.

Referee #1: Minor comments: Line 7: Droughts can also be temperature-driven without a decrease in precipitation. I also would consider it an overstatement that droughts belong to the most dangerous natural hazards.

*Response: We will rephrase this sentence.

Referee #1: Minor comments: Lines 40–42: Add citations to Seftigen et al. 2017, Helama et al. (2009) and Ljungqvist et al. (2019, ERL) here.

*Response: We will add these references.

---

## Author Response (AR1)

**Response to anonymous referee #1:**

*Many thanks for your kind and helpful comments, we appreciate them very much.*

Referee #1: 1) The authors are missing a number of recent important studies regarding historical droughts in Europe. While they are pretty well aware of the research in historical climatology, they are missing works in other fields of high-resolution palaeoclimate science.

*Response: Many thanks for this useful and necessary comment. We added the following publications: Helama et al., 2009; Seftigen et al., 2017; Ljungqvist et al., 2019; Metzger and Jacob-Rousseau, 2020.*

Referee #1: 2) I am lacking a quantitative comparison between the new documentary-based drought reconstructions and the tree-ring based Old World Drought Atlas. Without such a quantitative comparison it is very hard – or even impossible – to really know the nature of the similarities and disagreements.

*Response: We added such a comparison by using Pearson correlation coefficients.*

Referee #1: 3) Several of the figures are totally unreadable (and unpublishable). For example, the years are overlapping with each other so it cannot be read: much longer increments (say, 25 years) are needed for a clearer visualization. I also recommend the authors to look at articles in Climate of the Past for getting inspiration how to improve the graphs.

*Response: The figures have been modified.*

Referee #1: Minor comments: Line 7: Droughts can also be temperature-driven without a decrease in precipitation. I also would consider it an overstatement that droughts belong to the most dangerous natural hazards. *Response: The sentence has been rephrased.*

Lines 40–42: Add citations to Seftigen et al. 2017, Helama et al. (2009) and Ljungqvist et al. (2019, ERL) here. *Response: Done.*

**Response to anonymous referee #2:**

*Many thanks for your kind and helpful comments, we appreciate them very much.*

Referee #2: 1) I miss the general analysis and direct comparison of the two series in the Results chapter. Here only the greatest extremes are addressed and – despite the fact that the authors have made significant efforts to build up large-scale comparative series from Western Europe – only the greatest large-scale extremes are addressed in brief. Here, I think, a proper comparative analysis should be added: how different the individual series are from the regional series, what could be the reasons (e.g. source availability? Scale of events? Differences in the intensities or impacts on societies? etc.) for these differences (and the similarities).

*\* Response: We removed the analyses from the discussion chapter to the results and added more analyses (see also response to referee #1).*

Referee #2: 2) The authors provide direct comparisons of the evidence derived from the two cities in the Discussion chapter. As the entire paper (so as the title) is mainly concentrated on this topic, in my opinion this part of the Discussion chapter should be moved to the Results chapter.

*Response: Done.*

Referee #2: 3) I think the Discussion chapter could be organised somewhat differently: here smaller but important specific topics could be discussed. I think the discussion of 1473 and 1540, and its different appearance in the two cities are a good idea for one topic (i.e. I would keep it there), but plenty of other important questions could be addressed here. For example, the authors refer to the tree-ring based OWDA as one of the applied databases in the paper: in the Discussion chapter the authors could e.g. systematically compare the OWDA with the documentary evidence and list similarities and potential differences. Other possibility could be, for example, the discussion of uncertainties.

*Response: We removed parts of the discussion chapter to the results and added new information concerning a quantitative comparison of the OWDA with the DIB, DIR and SDI to the discussion chapter. We also added a paragraph on the uncertainty of dating.*

Referee #2: Minor points: 1) It is rather remarkable that in Rouen only the droughts prior to summer could be detected. In the paper the authors explain this situation with source availability.
I was just wondering: is it possible that for such a large town as Rouen no source exist at all that describe any other part of the year that contains any weather-related information? No any weather(-related) information at all in narratives, no other institutional documentation (e.g. municipal accounts)? It is rather unusual, especially with regards to the later part of the study period – and if this is the case, I think, should be more emphasised in the paper – already in the Source description part. The authors explain this phenomenon mainly with the difference in documentation practice.
*We wanted to use one continuous series of sources for each city, therefore we examined the proceedings of the city council. For Bern, there are much too many city accounts in order to read all of them, in the case of Rouen there are too many gaps. Therefore, we decided to analyse the proceedings. In order to get more information, also other documentary data were used, but those do not cover the whole period. We added more information about this decision to the data chapter.*
However, documentation practice is always related to 2) Is it really the case that in the documentation of the two cities only and exclusively the great and extreme dry conditions are mentioned, and never even moderate dry conditions? It is true that, usually, references on moderate dry conditions in European documentation are less pronounced. Still, they appear in documentation. Thus, it is a rather interesting and unique fact that, as the authors suggested, in neither of the two cities any "dry conditions" (i.e. without referring to any extreme) have been mentioned. Or, do you mean that all cases when "dry conditions" were mentioned had to be great (i.e. no. 2) or extreme (no. 3) droughts? It is really just a question out of curiosity. The question is also addressed to understand better the level of potential uncertainties of the index values.
*In the index -1 means a tendency to dry weather conditions. We did not consider this as droughts and therefore these index values are not included into the drought reconstruction. We added a few words about this.*
Referee #2: Minor point: 3) The authors suggest that mainly spring and summer droughts could be detected. This is a typical characteristics in Western and Central Europe (actually, also in Eastern Europe). Does this mean that the authors found no autumn and winter drought mention at all? Or did you find some? Because if you did, it would be perhaps also interesting for a short discussion in the "Discussion" chapter. Maybe not – it is up to you (just for a further potential idea into the Discussion chapter).
*Yes, we found a few autumn and winter droughts. We added some more words in order to make this point clearer.*

[revised manuscript text omitted]